# Phytocompounds as an Alternative Antimicrobial Approach in Aquaculture

**DOI:** 10.3390/antibiotics11040469

**Published:** 2022-03-31

**Authors:** Naqiuddin Nik Mohamad Nek Rahimi, Ikhsan Natrah, Jiun-Yan Loh, Francis Kumar Ervin Ranzil, Madi Gina, Swee-Hua Erin Lim, Kok-Song Lai, Chou-Min Chong

**Affiliations:** 1Aquatic Animal Health and Therapeutics Laboratory, Institute of Bioscience, University Putra Malaysia, Serdang 43400, Selangor, Malaysia; gs61890@student.upm.edu.my (N.N.M.N.R.); natrah@upm.edu.my (I.N.); 2Department of Aquaculture, Faculty of Agriculture, University Putra Malaysia, Serdang 43400, Selangor, Malaysia; 3Centre of Research for Advanced Aquaculture (CORAA), UCSI University, Cheras 56000, Kuala Lumpur, Malaysia; lohjy@ucsiuniversity.edu.my (J.-Y.L.); ervin_francis@outlook.com (F.K.E.R.); 4Health Sciences Division, Abu Dhabi Women’s College, Higher Colleges of Technology, Abu Dhabi 41012, United Arab Emirates; gmadi@hct.ac.ae (M.G.); lerin@hct.ac.ae (S.-H.E.L.); lkoksong@hct.ac.ae (K.-S.L.)

**Keywords:** aquaculture, herbal products, chemical agents, natural remedies, phytocompounds

## Abstract

Despite culturing the fastest-growing animal in animal husbandry, fish farmers are often adversely economically affected by pathogenic disease outbreaks across the world. Although there are available solutions such as the application of antibiotics to mitigate this phenomenon, the excessive and injudicious use of antibiotics has brought with it major concerns to the community at large, mainly due to the rapid development of resistant bacteria. At present, the use of natural compounds such as phytocompounds that can be an alternative to antibiotics is being explored to address the issue of antimicrobial resistance (AMR). These phytocompounds are bioactive agents that can be found in many species of plants and hold much potential. In this review, we will discuss phytocompounds extracted from plants that have been evidenced to contain antimicrobial, antifungal, antiviral and antiparasitic activities. Further, it has also been found that compounds such as terpenes, phenolics, saponins and alkaloids can be beneficial to the aquaculture industry when applied. This review will focus mainly on compounds that have been identified between 2000 and 2021. It is hoped this review will shed light on promising phytocompounds that can potentially and effectively mitigate AMR.

## 1. Introduction

There has been increasing interest in the investigation of different species of plants to identify their potential therapeutic applications as medicines. Due to their easy availability, cost effectiveness, presumed safety and biological friendliness, plant-based therapeutics or phytotherapeutics are very much preferred over synthetic molecules, which may not only be limited to the pharmaceutical sector but also found in the aquaculture sector [1].

Aquaculture is classified as an agricultural activity that is growing rapidly worldwide to address food security and to promote more sustainability efforts in food production [2]. In 2021, the market size value of global aquaculture production was USD 267,423.64 million in the global market. It is expected that the market size will increase over the years as the projected market size has been valued at USD 357,903.27 million by 2028 [3]. As the world’s largest aquaculture producer, China’s aquaculture production output currently meets approximately one-third of the human consumption of all capture and aquaculture fishery products worldwide [4]. According to a recent report by the FAO [5], over 91% of global aquaculture production is presently being produced in the Asian region (103 million tons in 2017), and the total global aquaculture production now exceeds that of global capture fisheries by over 18.32 million tons. To better meet the growing supply of aquaculture products, the aquaculture sector will need to develop strategies to ensure a consistent supply of aquatic offspring, in addition to maintaining quantity and quality [6].

The emergence of infectious diseases is usually triggered by ecological changes, often associated with human interventions, such as the transfer of organisms, environmental degradation, agricultural practices or technology [7,8,9,10,11,12]. Climate change and commercialized fish farming may have also contributed towards an inconsistent ratio of pathogen–host and environment interactions, with novel pathogens being observed and/or isolated annually in addition to existing diseases emerging in different geographical regions and species [13]. Disease outbreaks have affected the production of aquaculture with an annual loss of more than USD 6 billion [14]. To prevent bacterial diseases, tons of antibiotics, e.g., enrofloxacin, oxytetracycline and florfenicol [15], have been incorporated in fish feeds globally. Although the use of chemotherapeutic drugs is practical and easy to implement in aquaculture, the overuse or continuous use of antibiotics in aquaculture health management has resulted in the emergence of drug-resistant genes and multiple antibiotic resistance (MAR) bacteria in the aquatic environment of fish and shellfish [16]. Antimicrobial use exerts selective pressures which give rise to antimicrobial resistance. Despite the efficacy and advantages offered by chemical-based drugs, their application not only causes destruction to the environment but also poses a health risk to humans when consumed [17]. 

Despite its benefits, the use of antibiotics in aquaculture systems can create serious economic and health problems [18]. Antibiotic residues have been found in several aquatic products from Vietnam and other Asian countries [19,20,21]. The World Health Organization has labeled antimicrobial resistance as a “serious threat to global public health that requires action across all government sectors and society” [22]. The Centers for Disease Control and Prevention reported that, in 2013, two million people were infected with bacteria resistant to at least one antibiotic agent commonly used in the United States; in addition, 2000 people died as a result of antibiotic-resistant bacteria [23]. In Europe, 400,000 people were infected with multidrug-resistant bacteria, which caused about 25,000 deaths, in 2007 [24]. Emerging incidences of antimicrobial-resistant pathogens in animal production increase treatment failure rates, besides undermining sustainable food animal production during disease outbreaks while possibly compromising animal welfare when antibiotic levels need to be increased [25]. Furthermore, chemicals such as heavy metal-based disinfectants used in aquaculture may also enhance antibiotic resistance in the environment [26]. 

The aquaculture supply chain may be an undermined route for transmission of antimicrobial-resistant bacteria, despite the phenomenon of horizontal gene transfer of resistance genes from cultured aquaculture species and their environment to humans [27]. Aquatic ecosystems are hydrodynamically connected and open. Microbes can be transmitted passively through the water column more than 50 km [28] and inhabit a new ecosystem [29]. Thus, antimicrobial resistance induced by aquaculture practice can be transferred to the clinically relevant microbial strains of hydrodynamically connected ecosystems [30]. A study demonstrated that aquaculture may favor the transmission and persistence of antibiotic resistance genes in the riverine ecosystems along the Mekong Delta [31]. Hence, it is important to use natural alternatives such as phytocompounds for the treatment of microbial diseases in aquaculture. Plant extracts have been known to improve the immune system of animals and humans, showing great prospects for application in aquaculture [32]. However, there is limited information on the bioavailability of compounds present as natural remedies in fish organs [33].

In this review, a compilation of recent studies on the antimicrobial potentials, extraction methods and application of phytocompounds as alternatives to the usage of drugs in aquaculture is interpreted and discussed, with an emphasis on scholarly works from the period 2000 to 2021. 

## 2. Phytocompounds

### 2.1. Flavonoids

Flavonoids consist of a large group of polyphenolic compounds having a benzo-*γ*-pyrone structure and are ubiquitously present in plants; they are synthesized by the phenylpropanoid pathway [34]. A recent report estimated that over 9000 different flavonoids have been identified to date throughout the plant kingdom [35], and at least several hundred which occur are edible components [36]. Due to their chemical structure, flavonoids (2-phenyl-benzo-γpyrone derivatives) are divided into flavanones, flavonols, flavones, isoflavones, flavonols and anthocyanins. According to one study, the structure of flavonoids contains a flavan backbone formed of two benzene rings (rings A and B) connected by a heterocyclic ring of pyron or pyran (ring C) (Figure 1) [37]. The classification of flavonoid compounds consists of the presence of a carbonyl group at the fourth carbon atom of the C ring, a double bond between the second and third carbon atoms in this ring and a number of hydroxyl groups or other groups. All naturally occurring flavonoids have three hydroxyl groups: two in ring A (positions 5 and 7) and one in ring B (position 3) [38].

Based on one study, the underlying structure of flavonoids can be subdivided into six major subclasses, namely, flavonols, flavones, flavanones, isoflavones, flavan-3-ols and anthocyanins [39]. Table 1 shows the major classes of flavonoids, their general structure and major sources from where they can be obtained.

### 2.2. Alkaloids

Alkaloids are a class of naturally occurring organic nitrogen-containing bases. Alkaloids have diverse and important physiological effects on humans and other animals [54]. They are isolated from plants; however, they have also been found in animals, insects, marine invertebrates and some microorganisms [55]. Examples of alkaloids include morphine, sparteine, quinine, ephedrine and nicotine. According to a report, more than 40,000 compounds of alkaloids have been identified [56]. Table 2 below shows alkaloids obtained from plants with their origins. 

### 2.3. Phenolic Acids

Phenolic acids are described as phenolic compounds having one carboxylic acid group, and they are one of the main classes of plant phenolic compounds. Unlike flavonoids, free phenolic acids, such as benzoic, phenylacetic and cinnamic acids, have good bioavailability and good water solubility. Phenolic acids are a specific class of polyphenols that are usually involved in mechanisms of defense against biotic and abiotic stresses [62]. Phenolic acids have substantial lipid and water solubility; they have been shown to inhibit oxidative deterioration when added as functional ingredients in emulsion model systems [63]. They can be absorbed in the stomach, while flavonoids cannot be absorbed, and only a small amount of flavonoids are transported passively through the intestinal wall into the blood [64]. Examples of phenolic acids are gallic acid, caffeic acid, rosmarinic acid and carnosic acid (Table 3). 

### 2.4. Terpenoids

Terpenoids, also known as isoprenoids or terpenes, are a large class of natural products found in nearly all living organisms [69]. There are about 60,000 terpenoid structures that have been identified, making them one of the largest classes of natural products known [70]. Terpenoids can be found abundantly in plants compared to other living organisms. A report estimated that the number of distinct terpenoid compounds (an inclusive term used to describe both terpenes and compounds with terpene moieties linked to other moieties derived from different pathways) in plants could be in the scores of thousands [71]. A recent report noted that many plants containing terpenoids have been used widely in traditional medicine as they have high anti-inflammatory and pain-relieving properties [72]. The lineage-specific terpenoids, which have arisen throughout the evolution of green plants, have generally been postulated to play a role in the ecological interactions of plants with biotic and abiotic aspects of their environment [73]. Such roles have included defense against herbivores and pathogens, including signals and rewards for beneficial organisms, such as pollinators and mycorrhiza [74]. Table 4 below shows the classification and structure of terpenoids with their origins.

### 2.5. Saponins

Saponins are a class of bioorganic compounds. More specifically, they are naturally occurring glycosides described by their soap-like foaming property; consequently, they produce foams when shaken in aqueous solutions [79]. Saponins have one or more hydrophilic glycoside sugar moieties combined with a lipophilic triterpene molecule [80]. Scientific research reveals that saponins display medicinal properties such as anti-inflammatory [81], antidiabetic [82] and antibacterial effects [83] and play a role in cytotoxic activity against tumor cells [84]. As a result, saponins have the potential to provide a platform for the development of drugs based on natural products [85]. Table 5 below show examples of plants that are rich in saponins.

## 3. Extraction Methods of Phytochemicals from Plants

Phytochemicals, also referred to as phytobiotics or phytogenics, are natural bioactive compounds that are derived from plants and incorporated into animal feed to enhance productivity [89]. Phytochemicals can be used in solid, dried and ground forms or as extracts (crude or concentrated) and can also be classified as essential oils (EOs). EOs are volatile lipophilic substances (obtained by cold extraction or steam/alcohol distillation) and oleoresins (extracts derived using non-aqueous solvents), depending on the process used to derive the active ingredients [90]. There are benefits of such compounds in plants, and researchers must use and know the appropriate extraction techniques to maximize yield and quality [91].

### 3.1. Extraction Process

Processing is an important step for the use of herbal medicines in traditional Chinese medicine [92]. There are many extraction techniques including conventional ways such as maceration, Soxhlet extraction, percolation, reflux extraction and distillation; these have all been used to obtain extracts from plants [93]. The disadvantages of these methods include longer extraction times, the use of expensive and high-purity solvents, lower extraction selectivity and thermal decomposition of heat-labile compounds at higher temperatures [94].

For the best extraction method, extraction should be realized in a short time and with minimal solvent consumption because a short extraction time reduces power consumption and possible decomposition of the active components [95]. Table 6 shows the solvents used for the extraction of phytochemical components. The extraction process can be divided into two categories, which are the traditional extraction methods and newly emerging technologies based on energy or mechanisms [96]. New and promising techniques have been introduced such as ultrasound-assisted extraction, microwave-assisted extraction (MAE), enzyme-assisted extraction and supercritical fluid extraction [97].

#### 3.1.1. Conventional Extraction Method

##### Soxhlet Method

One of the traditional methods of extraction is Soxhlet extraction. Soxhlet extraction is the most commonly used method for extracting phenolic compounds due to its lower processing cost, simplicity of operation, suitability for both initial and bulk extraction, and for the total recovery of extracts, lower time consumption and solvent options as compared to other conventional methods such as maceration or percolation [98]. Table 7 shows a comparison of the Soxhlet method and various extraction methods for natural products. However, the disadvantage of the Soxhlet method is that the process is very time-consuming and requires a large amount of solvents [99]. This method does not require the separation of the extraction result, unlike the other extraction methods.

##### Maceration

Maceration is not only a simple but also the cheapest conventional method because it only requires a container as the place for extraction; this method, however, requires a long time for the extraction process [100]. The process takes place only by molecular diffusion which needs sufficient time for the solvent to be allowed to pass through the cell wall and solubilize the constituent present in the plant [101]. Maceration involves soaking plant materials (coarse or powdered) in a sealed container with a solvent and keeping them at room temperature for at least three days with frequent stirring [102]. The advantages of maceration are that it does not require any special equipment or a skill-based operator, and it is an energy-saving process. Unfortunately, it comes with a drawback as the process takes up to a few weeks [101].

##### Percolation

Percolation is more efficient than maceration because it is a continuous process in which the saturated solvent is constantly replaced by a fresh solvent [103]. Raw materials are moisturized with the menstruum for a period of 4 h in a separate closed vessel; this process is called imbibition [104]. An additional solvent is added to form a shallow layer above the mass, and the mixture is allowed to macerate in the closed percolator (a narrow, cone-shaped vessel open at both ends) for 24 h [101]. An additional solvent is added as required, until the percolate measures about three-quarters of the required volume of the finished product [102]. The extract is then pressed, and the liquid is added to the percolate. A sufficient amount of solvent is added to produce the required volume, and the mixed liquid is clarified by filtration or by standing followed by decanting. The process is repeated until a drop of the solvent from the percolator does not leave a residue when evaporated.

##### Decoction

Decoction is a process that involves continuous hot extraction using a specified volume of water as a solvent [105]. Dried, ground and powdered plant materials are placed into a clean container. Water is then poured and stirred. Heat is then applied throughout the process to hasten the extraction [106]. The process usually only takes 15 min. Decoction is the method of choice when working with tough and fibrous plants, barks and roots, and also with plants that have water-soluble chemicals [101]. This process, however, cannot be applied for the extraction of thermolabile or volatile components [103].

#### 3.1.2. Advanced Extraction Method

In recent decades, new and more rapid efficient extraction methodologies have been developed which minimize the use of solvents and shorten extraction times while maximizing extraction yields and preventing the risk of degrading the compounds of interest, which is the main goal of extraction [93]. There are many advanced techniques such as microwave-assisted extraction (MAE), ultrasound-assisted extraction (UAE), pressurized liquid extraction (PLE) and supercritical fluid extraction. Modern methods such as microwaves and ultrasonication facilitate efficient extraction due to their fast removal of cuticular superficial waxes from plants [107].

##### Microwave-Assisted Extraction (MAE)

This technique is based on nonionizing electromagnetic radiation, which moves in the form of sinusoidal waves [108]. Radiation causes rotation of the dipoles of the molecules and the migration of ions, disturbing the hydrogen bonds of the food components. The ions maintain their direction according to the signals of the electric field generated by the waves. The field oscillates, and therefore the directions of the ions change frequently, sliding from one side to another and/or rotating on themselves, which causes friction and collision between them, generating heat [109]. A study reported that there are two types of MAE methods: solvent-free extraction (usually for volatile compounds) and solvent extraction (usually for non-volatile compounds) [110]. A microwave that works by assisting electromagnetic waves makes the extraction time faster, as it only takes a few minutes. This is because all electromagnetic waves generated are converted directly into heat [111].

For example, a study concluded that the MAE technique returned a better yield of extraction compared to heat reflux extraction (HRE) and had a better extraction time compared to maceration and HRE [91]. Another study also reported that the best method for extracting the active compound from soursop leaves was the MAE technique, with a yield of 33.98% compared to maceration (10%) and the Soxhlet method (29.13%) [112]. MAE is also regarded as a green technology because it reduces the usage of organic solvents [103]. A study on the extraction of catechin from *Arbutus unedo* fruits comparing the maceration, microwave-assisted extraction and ultrasound extraction techniques showed that microwave-assisted extraction (MAE) was the most effective, since a lower temperature was applied in maceration with nearly identical extraction yields to MAE, which can be translated into economic benefits [113].

### 3.2. Importance of Extraction of Phytocompounds

The extraction process of the bioactive compounds from plants depends on the type of extraction process that will be applied along with the parameters such as solvents, polarity, temperature and pressure [114]. These factors influence the types of metabolites found predominantly in the crude extract. The type of solvents used for extraction can be categorized based on polarity, ranging from non-polar solvents such as hexane and trichloromethane to the highly polar solvent of water [115,116]. A study found that methanol, water and acetone extracts yielded positive results for many groups of compounds in a phytochemical screening test [117]. Biscaia et al. [118] reported that the extraction yields of methanol extracts were higher than other solvent extracts with a decrease in polarity, which indicated that most of the soluble components in seaweeds were high in polarity. It was found that the use of ethanol as a co-solvent in SFE increased the extraction yield of propolis by about three times compared to CO_2_ extraction. It was also stated that the pressure difference of 100, 150, 200 and 250 bar showed a significant increase in the extraction yield, with 6.4 ± 0.4, 9.0 ± 0.1, 10 ± 1 and 12 ± 1% *w*/*w*, respectively. Hence, it is important to use various methods in extraction according to the suitability of the plant.

**Table 7 antibiotics-11-00469-t007:** Comparison of extraction methods and their corresponding phytocompounds.

Method	Solvent	Temperature	Duration	Compound(s) Extracted	Advantages	Disadvantages	References
Maceration	Water, aqueous and non-aqueous solvents	28–30 °C	3–4 days	Phenolics, flavonoids and alkaloids	Cheap, with no special tools required and less energy processes.	Time-consuming and high solvent usage.	[119]
Percolation	Water, aqueous and non-aqueous solvents	25–40 °C	24 h	Phenolics and flavonoids	Shorter time than maceration and is possible to extract thermolabile constituents.	Needs skill and takes longer time than Soxhlet extraction.	[120]
Soxhlet extraction	Organic solvents	<60 °C	16–20 h	Andrographolide and deoxyandrographolide	Able to extract large sample materials, less skill required and solvent savings.	High risk of thermal destruction of compounds and time-consuming.	[121]
Decoction	Water	70 °C	0.5–1 h	Catechoo-tannins, anthraquinones, phenolics and alkaloids	Suitable for heat-stable compounds and less skill required.	Unsuitable for heat-sensitive compounds.	[122]
Microwave-assisted extraction	Water, aqueous and non-aqueous solvents	70–80 °C	3–5 min	Phenolics, alkaloids and carotenoids	Less organic solvents are needed, high extraction rate and no airborne contamination.	Limited amount of sample that can be extracted.	[123]

## 4. Phytocompounds as Alternatives to Antimicrobial Approach in Aquaculture

### 4.1. Antimicrobial Activities in Aquaculture

In this decade, the development of scientific research has continuously increased the discovery of phytochemicals from plants (Table 8). Furthermore, the use of plant extracts in the traditional extraction method in aquaculture is considered to be an ecofriendly approach and cost-effective method, which does not cause side effects [124].

#### 4.1.1. Antibacterial Activity

In general, bacteria can be classified based on their cell wall structure, whether it is Gram-positive or Gram-negative [143]. Unlike viruses, which are transported by airborne migration, most bacteria achieve motility (in a fluid environment) by flagella which are characteristically nanomotors powered by ionized hydrogen or protons (H^+^) [144]. The antimicrobial activity of various plants has been tested against many Gram-positive and Gram-negative bacteria by different research groups worldwide, highlighting the importance of herbs as alternative therapeutic agents in aquaculture [145]. The discovery of antimicrobial activities in plants through recent research has shown positive outcomes in eliminating such bacterial infections. Quercetin, a type of lipophilic compound, is important to human health and the development of functional foods [146]; it can also be naturally found in many foods. Kim et al. [147] found that a quercetin-pivaloxymethyl conjugate (Q-POM) at 5 μg/mL inhibited 70% of biofilm establishment by a vancomycin-resistant *Enterococcus faecium* isolate.

Another example of a compound that exhibited antibacterial activity is kaempferol 3-*O*-α-L-(2″,3″-di-*Z*-*p*-coumaroyl)rhamnoside isolated from *Platanus occidentalis*. The compound exhibited high antibacterial efficacy against MRSA (IC_50_ 0.4 mg/L) and *Streptococcus iniae* LA94-426 [129]. Alkaloids such as koumine, gelsemine and gelsenicine have been demonstrated to possess antibacterial effects. Ye et al. [127] showed that Wuchang bream (*Megalobrama amblycephala*) fed with 20 mg/kg or 40 mg/kg of *Gelsemium elegans* alkaloids possessed a high survival rate upon *Aeromonas hydrophila* challenge as well as a higher growth performance as compared to that of the negative control. Furthermore, no alkaloid residues were detected in the orally administrated fish [127]. Hence, they can be used as efficient and environmentally friendly additives in aquaculture. On the other hand, the phytocompound lycorine has been reported to have antibacterial effects against a fish bacterial pathogen, *Flavobacterium columnare* [148]. The phytocompound gallic acid found in *Eucalyptus globulus* showed effective antibacterial activity against seven fish pathogenic bacteria when tested using a growth inhibition test [149].

In a recent study, fishes that were exposed to 1µg/L of quercetin via the immersion method with different concentrations of 0.01, 0.1, 1, 10, 100 and 1000 μg/L significantly showed the highest acid phosphatase (ACP) and myeloperoxidase (MPO) activities and complement (C3) contents in *Danio rerio* compared to other concentrations [150]. Other examples of phytocompounds are 17-pentatriacontene, octasiloxane and stigmasterol that were extracted from *Eichhornia crassipes*, which increased bacterial resistance in *Channa punctate* against *Vibrio harveyi* infection [126]. The plant species *Chelidonium majus* has phytocompounds such as chelerythrine chloride that showed strong toxicity against *Edwarsiella ictalurid*, with a 24 h LC of 7.3 ± 0.8 mg/L and MIC of 2.1 ± 1.7 mg/L [125]. A report also stated that *Macleaya cordata* that contained sanguanarine at two concentrations (1 and 1.5 mg/kg of feed) improved the survival rate and resistance to *Vibrio parahaemolyticus* infection of *Litopenaeus vannamei* [128].

Some phytocompounds possess no bactericidal properties but can disrupt the quorum sensing of bacteria. Quorum sensing is a bacterial cell-to-cell communication system that allows bacteria to share information about fluctuations in cell population density and adjust gene expression accordingly to give rise to the most beneficial phenotypes [151]. Plant-derived compounds such as tocopherol and phytol have been shown to interrupt the quorum sensing of aquatic pathogenic microbes. The quorum quenching properties of tocopherol and phytol have been demonstrated to significantly decrease the production of biofilm and virulence factors of *Vibrio campbellii*, as well as increasing the survival rates of infected tomato clownfish [152]. The terpene thymol showed the strongest antimicrobial activity against *Streptococcus aureus*, and it possessed the highest capacity to increase the permeability of PC LUVs, together with the ability to migrate across an aqueous medium and to interact with phospholipidic membranes [153]. Further, Cristani et al. [153] found that monoterpenes such as p-cymene and carvacrol are more active against the Gram-negative *Escherichia coli*; their calorimetric behavior shows that they markedly affect the membrane lipid composition, taking the place of lipid molecules, and are strongly absorbed by lipidic membranes, so that they are not released and transferred to other bilayers. In addition, the detection of the terpene-like molecule S-12-hydroxyfarnesyl-L-cysteine indicated that *Chlorella* sp. may secrete bioactive molecules as a defense mechanism [154]. Moreover, catechin, a flavonoid-like compound extracted from the bark and leaves of the plant *Combretum albiflorum,* has been described to inhibit the production of violacein in *C. violaceum* CV026 and pyocyanin in *P. aeruginosa* PAO1 [155].

#### 4.1.2. Antiparasitic Activity

Parasites represent one of the most successful modes of life in nature [156] that have arisen on multiple independent occasions in many phyla [157]. They are a natural component of the environment and may be viewed as an indicator of the relative health of an ecosystem [158]. Most parasite species rarely cause problems in the natural environment, but in aquaculture, parasites often cause serious outbreaks of disease [159]. Most of the commonly encountered fish parasites are protozoans. They are single-celled organisms, many of which are free-living in the aquatic environment [160]. The obvious effect of these parasites is irritation of the epithelial surface, causing the fish to stress [161]. Treatment with medicinal plants having antimicrobial activity is a potentially beneficial alternative in aquaculture.

Zheng et al. [130] reported that phytocompounds such as gracilin and zingibernsis newsaponin found in *Costus speciosus* can kill 100% of *Ichthyophthirius multifilis* by in vitro treatments. The EC_50_ values were recorded to be 0.53 and 3.2 mg/L, respectively. It was reported by Zhang et al. [131] that *Galla chinensis* contains pentagalloylglucose. This phytocompound was responsible for eliminating all *Ichthyophthirius multifiliis* theronts at concentrations of 2.5–20 mg/L, and for completely interferring with the reproduction of tomonts at 40 mg/L. A 93.3% rate of survival was achieved in the Ich-infected catfish treated with pentagalloylglucose at 20 mg/L, whereas all infected fish were dead in the negative control group. Other than that, phytocompounds such as sanguinarine, β-allocryptopine and 6-methoxyl-dihydro-chelerythrine that were extracted from *Macleaya microparpa* showed potent anthelmintic activity against *Dactylogyrus intermedius* in *Carassius* auratus, with EC_50_ values of 0.37, 4.64 and 3.63 mg L^−1^, respectively [132]. It was also reported that *Macleaya microparpa* contains dihydrosanguinarine and dihydrochelerythrin. The EC_50_ values of these compounds against *I. multifiliis* were 5.18 and 9.43 mg/L, respectively [133]. Finally, Zhou et al. [134] reported that *Polygonum cuspidatum* contains emodin. In vitro treatment of the compound at 1 mg/L took 96 min to kill all *I. multifiliis.* The recovery of infected *Ctenopharyngodon idella* can be achieved by continuously adding emodin for 10 days.

#### 4.1.3. Antiviral Activity

Viruses are obligate intracellular parasites, meaning that they are completely dependent upon the internal environment of the cell to create new infectious virus particles, or virions [162]. The infectious form of a virus, the virus particle or virion, replicates itself by entering a host cell, disassembling itself and copying its components, which are then assembled into progeny virus particles [163].

Extracts from mangrove plants and associates have been used worldwide for medicinal purposes, and having around 349 metabolites recorded, they turn out to be a rich source of steroids, diterpenes and triterpenes, saponins, flavonoids, alkaloids and tannins [164]. Sudheer et al. [165] found that extracts from *Rhizophora mucronata*, *Sonneratia* sp. and *Ceriops tagal* were found to have virucidal properties against white spot syndrome virus (WSSV). Similarly, *Rhus verniciflua* contained the phytocompounds fisetin, fustin and sulfurustin that possessed antiviral activity. Fisetin showed the highest significant anti-infectious activities against hemorrhagic necrosis virus and antiviral hemorrhagic septicemia virus, showing EC_50_ values of 27.1 and 33.3 µM. Fustin and sulfuretin displayed significant antiviral activities, showing EC_50_ values of 91.2–197.3 μM against IHNV and VHSV [139]. The compound polyhydroxy isocopalane, a terpene, was extracted from the marine sponge *Callyspongia* sp., and the compound was found to exhibit strong antiviral activity against WSSV at a concentration of 60 mg/L and resulted in 34% survival of the WSSV-infected *Litopenaeus vannamei*. The presence of the –OH group in polyhydroxy isocopalane was found to be responsible for inhibiting the replication of the DNA virus and amino acids on the active sites in the host organism cell [166].

Moreover, research by Qian and Zhu [167] showed that hesperetin exhibits vital effects on the protection of *Procambarus clarkia* against white spot syndrome virus. Another example of a phytocompound with antiviral activity is quercetin, which is extracted from *Illicium verum*, a famous medicinal plant also known as a drug homologous food in China. Furthermore, quercetin (50 μg/mL) has the greatest antiviral activity, with a percent inhibition of 99.83%, making it a potential candidate for developing effective drugs for controlling SGIV infection [168]. The plant species *Polysiphonia morrowii* that contains 3-bromo-4, 5-dihydroxybenzy methyl ether exhibited significant antiviral activities, showing selective index values (SI = CC_50_/EC_50_) of 20 to 40 against IHNV and IPNV [137]. Further, the phytocompound gymnemagenol (3β, 16β, 28, 29-tetrahydroxyolean-12-ene) extracted from *Gymnema sylvestre* showed antiviral activity against WSSV. At 20 µg/mL of gymnemagenol, it inhibited 50% of cell viability of grouper nervous necrosis virus (GNNV), which showed effectiveness in inhibiting the proliferation of GNNV in infected SIGE cells [136].

#### 4.1.4. Antifungal Activity

Fungal infection is one of the foremost disease problems in shellfish and finfish aquaculture [169]. It was revealed that an opportunistic fungal pathogen was responsible for the mortality of broodstocks in aquaculture [170]. The majority of fungi which can cause infection in fish are opportunistic and not exclusive parasites of fish [171]. For example, chytridiomycosis is an infectious disease caused by the chytrid fungus *Batrachochytrium dendrobatidis*; it is a non-hyphal zoospore fungus that comes under the phylum Chytridiomycota [172] or Blastocladiomycota [173]. Baleta et al. [174] found that phytocompounds such as flavonoids, tannins, saponins, phenolics, sterols and terpenoids were obtained from *Sargassum oligocystum* and *Sargassum crassifolium* by ethanolic extracts. Ethanolic extracts demonstrated the phytochemical constituents responsible for the antimicrobial property. *Sargassum polycystum* and *Sargassum tenerrimum* showed significant activity against fungal pathogens [175]. Recent research revealed that the use of *Magnolia officinalis* and *Euphorbia fischeriana* shows significant antifungal effects against *Saprolegnia* sp. [176].

Effiong and Sanni [177] found that ethanolic extracts of tannins and steroids that were extracted from *Lemna pauciscostata* have antifungal properties against the mycelial growth of fish feed spoilage fungi such as *Fusarium oxysporium*, *Penicillium digiatum*, *Aspergillus niger*, *Aspergillus fumigatus*, *Aspergillus flavus*, *Rhizopus oryzae* and *Rhizopus stolonifera*. Other than those, compounds such as palmitic acid and squalene extracted from *Ulva lactuta* showcased a significant zone of inhibition of antifungal activity against *Aspergillus niger* (36 mm) when compared to other antibiotic drugs [142]. Moreover, *Zataria multiflora* extract containing p-cymene and carvacrol at a concentration of 25 ppm, *Eucalyptus camaldolensis* containing anethole and 1, 8-cineole at a concentration of 25 ppm and *Geranium herbarium* containing geraniol and dihydrogeraniol at a concentration of 100 ppm for 60 min daily with three repetitions were the best treatments in *Onchorynchus mykiss*, represented by the prevention of fungal attack, and the increase in the hatching rate, the eyed egg rate and the final larvae rate [140]. The compounds hexadec-2-en-1-ol and carvacrol found in *Thymus linearis* showed significant specificity towards the V-type ATPase site and TKL protein kinase of *Saprolegnia parasitica* responsible for the virulence of pathogens [141].

## 5. Challenges and Future Perspective

Despite numerous studies on the discovery and characterization of new plant-derived compounds being on the rise, their successful on-site farm application is limited and their effective performance is inconsistent [178]. In vivo studies are needed to understand the underlying mechanism and pharmacodynamics of phytocompounds as well as the safe dosage. Translation of in vitro tests into in vivo application is challenging in aquaculture. Some phytocompounds possess very close median effective concentration (EC_50_) and acute toxicity (LC_50_) values, thus making them dangerous when used by laymen as an overdose of these compounds tends to occur [133].

Aquaculture leverages the aspects of practicability and profitability. Instead of a purified phytocompound, part or whole of the therapeutic plants are used and administered to aquatic animals orally as feed additives due to the cheaper cost and high practicability [179]. This practice may pose a negative impact: for example, a high dietary fiber content stunts fish growth [180]. Additional steps such as fermentation have been proposed to reduce the fiber content of the medicinal plant before its incorporation into fish feed [180,181]. Extensive studies of various aspects including the preservation of bioactive phytocompounds, and fermentation conditions, are necessary for each medicinal plant of interest.

Furthermore, these phytocompounds have proved their potency when acting as antimicrobial agents against fish pathogens including bacteria, viruses and parasites. It is also important to use various extraction methods that are suitable for certain compounds as the amount of product compounds can vary. Most importantly, the phytocompounds extracted from plants are only produced in a small volume which can only be used for research purposes. It is necessary to overcome this problem, as further research needs to be conducted to synthesize the compounds for the aquaculture industry.

Phytocompounds have the prospect of becoming an antimicrobial agent alternative in aquaculture. Emerging technology and advanced research have showcased a wide range of potential activities in phytocompounds ranging from medium to high levels of antimicrobial activity towards freshwater and marine pathogens. Application of these phytocompounds would definitely be a step forward towards the usage of medicinal plants and herbs in aquaculture. However, further research on natural products will need to be conducted in the future not only to maximize the use of medicinal plants, but also to capitalize on the safety aspects of what phytocompounds can offer which may outweigh synthetically or semi-synthetically derived agents by far. Current technologies in disease detection together with effective treatment, prevention and control methods are integral to contain or slow down the spread of infectious diseases in cultured fish.

## Figures and Tables

**Figure 1 antibiotics-11-00469-f001:**
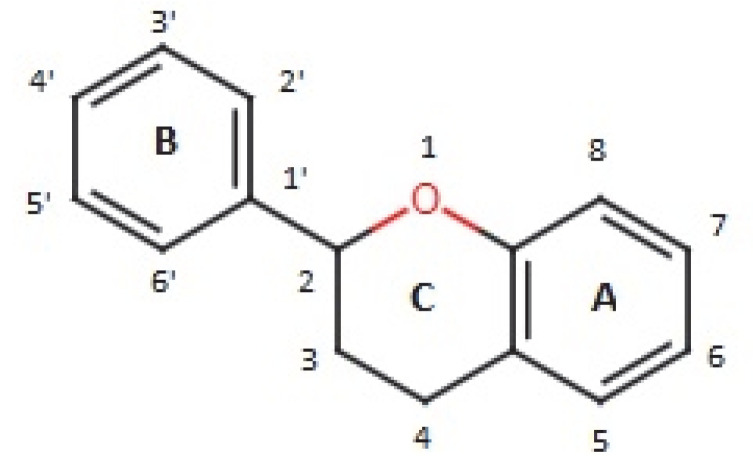
Structural characteristics are indicated by letters A, B and C; diphenylpropane skeleton.

**Table 1 antibiotics-11-00469-t001:** List of major classes and their phytocompounds with positive bioactivity from plants.

Class	General Structure	Phytocompound and Its Antimicrobial Properties	Plant Sources
Flavanol	** 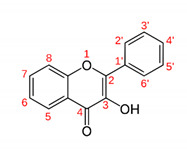 **	Catechin: able to inhibit the growth of methicillin-resistant *S. aureus* ATCC 33591 (MRSA) and methicillin-susceptible *Staphylococcus aureus* (MSSA, ATCC 25923).	*Anacardium occidentale* [40]
		Epicatechin gallate: epicatechin gallate enhanced the antibacterial effect of β-lactam antibiotics against MRSA in vitro and in vivo.	*Fructus crataegi* [41]
		Epicatechin: high efficacy of phytoisolate compound against the parasitic activity of *Paramphistomum cervi*.	*Ricinus communis* [42]
Flavonol	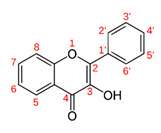	Kaempferol 3-*O*-α-L-(2″, 3″-di-*Z*-*p*-coumaroyl)rhamnoside: showed high efficacy against MRSA (IC_50_ 0.4 mg/L) and *Streptococcus iniae* LA94-426.	*Platanus occidentalis* [43]
		Myricetin 3′-glucoside and myricetin 3-alpha-L-arabinofuranoside: showed strong antiglycemic activity by inhibiting carbohydrate-hydrolyzing enzymes.	*Syzygium malaccense* [44]
		Quercetin 3-O-glucuronide: significant inhibitory effect of bacterial growth against *S. aureus*, *E. faecalis*, *E. coli*, *P. aeruginosa* and *Salmonella typhi* with inhibition zone diameters greater than 13 mm.	*Tamarix gallica* [45]
Flavone	** 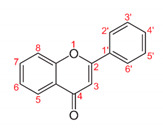 **	5-Hydroxy-3′,4′-dimethoxyflavone-7-O-(rhamnoside) and 5-hydroxy-3′-methoxyflavone-4′-O-(penthenyl-4-one)-7-O-(2″-(rhamnosyl) rhamnoside): able to inhibit the growth of *B. subtilis* (21.4 mm) and *E. faecalis* (8.2 mm) compared to tetracycline (22.2 mm and 9.6 mm, respectively).	*Achillea tenuifolia* [46]
		Apigenin: apigenin (10 µL) had antibacterial effects that were more significant on *S. typhimurium* and *P. mirabilis* when compared with streptomycin as a control (10 µL).	*Portulaca oleracea* L. [47]
Isoflavone	** 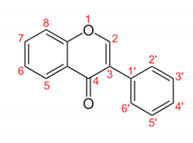 **	Genistein: Increased the acetylcholinesterase (AChE) activity and, in contrast, reduced both glutathione and catalase activity. The results may suggest beneficial impacts on cognitive defects related to Alzheimer’s disease.	*Glycine max* [48]
		Genistein: methanolic extracts containing genistein displayed antibiotic response against all bacterial strains and maximum zone of inhibition at a low concentration level at 350 µg/mL.	*Rhizophora apiculate* [49]
Flavanone	** 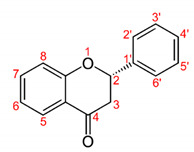 **	Hesperidin: able to inhibit the growth *Streptococcus aureus*, *Escherichia coli*, *Enterococcus faecalis* and *Pseudomonas auraginosa* at a 15% concenteration with inhibitory diameter range of 7.65 mm ± 0.36 mm to 9.96 mm ± 0.52 mm, and at a concentration of 20% with a diameter range of 9.26 mm ± 0.72 mm to 13.39 mm ± 028 mm.	*Citrus microparpa* [50]
		Hesperetin-A: showed a noteworthy cytotoxicity effect (IC_50_: 2.86 μg/mL) on HeLa cell line, and an in silico molecular docking study portrayed hesperetin as having a good interaction with the E6 protein of HPV16 cervical carcinoma, which is beneficial for cancer treatment.	*Cordia sebestena* [51]
Anthocyanidin	** 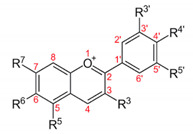 **	Pelargonidin: possessed potent scavenging activity for superoxide radicals to attract more neutrophils in plasma.	*Punica granatum* [52]
		Proanthocyanidins: exhibits anti-*Escherichia coli* adhesion activity with P-type fimbriae on the wall of the urinary tract.	*Vaccinium sect. Cyanococcus* [53]

**Table 2 antibiotics-11-00469-t002:** Plant alkaloid phytocompounds and their bioactive properties.

Alkaloids	General Structure	Phytocompound and Bioactive Properties	Plant Origin
Deoxytubulosine	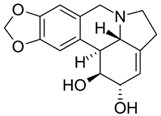	β-carboline-benzoquinolizidine alkaloid deoxytubulosine: exhibits cytotoxicity and anticancer activity against Dalton’s ascitic lymphoma cells.	*Alangium salvifolium* [57]
Carbazole	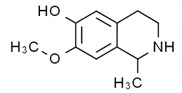	Methyl carbazole-3-carboxylate: showed the best in vitro cytotoxic activities against Hela, K562, A549, H1299 and SMMC-7721 tumor cell lines.	*Clausena lansium* [58]
Pyridazine	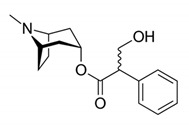	2,7-Diphenyl-1,6-dioxopyridazino[4,5:2,3]pyrrolo[4,5-d]pyridazine: showed high potency of antibacterial effects through inhibition zone against *Escherichia coli*, *Pseudomonas eurogenosa*, *Staphylococcus aureus*, *Proteus mirabilis* and *Klebsiella pneumonia.*	*Datura stramonium* [59]
Quinolizidine	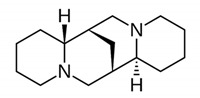	Lupanine, 13α-hydroxylupanine and albine: alkaloid extracts showed high antimicrobial activity against *K. pneumonia* and moderate activity against *P. aeruginosa* clinical isolates.	*Lupinus albus* [60]
Trigonelline	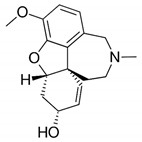	Trigonelline: at the dose of 1 g/L, it showed (1) an antihistamine effect on guinea pig ileum; (2) an anticholinergic effect on rat colon; (3) a stimulant effect on rat uterus.	*Trigonella foenum-graecum* [61]

**Table 3 antibiotics-11-00469-t003:** List of phenolic acids and their phytocompounds with positive bioactivity from plants.

Phenolic Acid	General Structure	Phytocompound and Bioactive Properties	Reference
Gallic acid	** 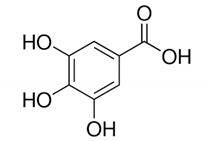 **	Gallic acid: showed a high zone of inhibition of 13.67 ± 0.58 mm towards *S. Aureus* through the disc diffusion method.	*Eucalyptus globulus* [65]
p-Coumaric acid	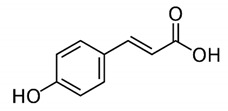	4-Hydroxycinnamic acid: exerted anti-inflammatory effects, in a mechanism that included suppression of inflammatory cell infiltration as well as the levels of tumor necrosis factor-α and interleukin 6.	*Oldenlandia diffusa* [66]
Rosmarinic acid	** 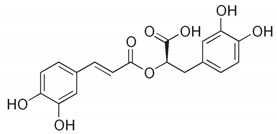 **	Rosmarinic acid methyl ester found in *Origanum vulgare* that possessed a strong antioxidant effect is much safer and less toxic than either arbutin or l-ascorbic acid in human fibroblast cells.	*Origanum vulgare* [67]
Ferulic acid	** 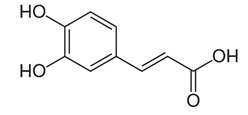 **	Trans-4-hydroxy-3-methoxycinnamicacid: inhibited UVB-induced matrix metalloproteinases that contribute to the development of skin cancer via post-translational mechanisms.	*Triticum aestivum* [68]

**Table 4 antibiotics-11-00469-t004:** List of common terpenoids and their phytocompounds with positive bioactivity from plants.

Terpenoids	Chemical Structure	Phytocompound	Plant Species
Sesquiterpenoids	** 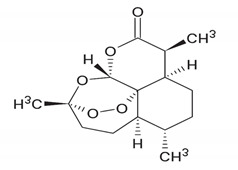 **	Artemisinin: acts as an inhibitor of the production of *Flaviviridae* viruses, and its effect is additive to interferon-α and ribavirin.	*Artemisia annua* [75]
Monoterpenoids	** 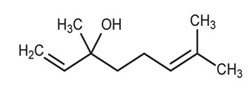 **	Linalool: responsible for the antipsoriatic activity of lavender oil as the compound showed more than 50% recovery in psoriasis area severity index scores and recovery level of Th-17 cell cytokines.	*Lavandula angustifolia* [76]
Triterpenoids	** 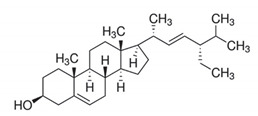 **	Stigmasterol: the compound exhibited 29 mm as the zone of inhibition against *Staphylococcus aureus*.	*Neocarya macrophylla* [77]
Diterpenoids	** 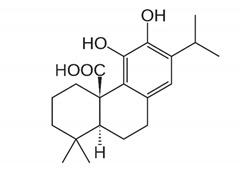 **	Carnosic acid and carnosol: exhibited a significant increase in antibacterial activity against *Listeria monocytogenes* and *Staphylococcus aureus* strains.	*Rosmarinus officinalis* [78]

**Table 5 antibiotics-11-00469-t005:** Plant-derived saponins and the phytocompounds with positive bioactivity derived from plants.

Saponins	Chemical Structure	Compound and Bioactive Properties	Plant Origin
Quinoa saponins	** 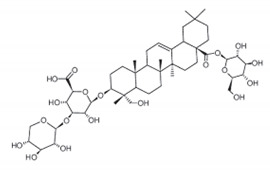 **	Compound exerted obvious bacteriostatic and bactericidal effects on Gram-positive bacteria such as *Staphylococcus aureus*, *Staphylococcus epidermidis* and *Bacillus cereus*.	*Chenopodium quinoa* [86]
Soyasaponin	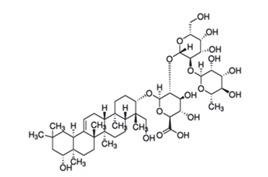	Soyasaponin Ab: colon shortening, myeloperoxidase activity, the expression of cyclooxygenase-2 (COX-2) and inducible nitric oxide synthase (iNOS) and activation of the transcription factor nuclear factor-kB (NF-kB).	*Glycine max* [87]
Ginsenosides	** 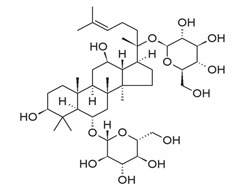 **	12-One-pseudoginsenoside F_11_: prevented H_2_O_2_-stimulated cell damage in A549 cells, which may be strongly related to the antioxidative effects of 12-one-PF11.	*Panax quinquefolium* [88]

**Table 6 antibiotics-11-00469-t006:** The solvents that are used for active phytochemical extraction.

Water	Ethanol	Methanol	Chloroform	Dichloromethanol	Ether	*n*-Hexane
Tannins Anthocyanins Terpenoids Saponins	Tannins Terpenoids Polyphenols Flavonoids Alkaloids	Phenolic acids Alkaloids Carotenoids Flavonoids	Flavonoids Terpenoids	Terpenoids	Alkaloids Terpenoids	Carotenoids

**Table 8 antibiotics-11-00469-t008:** Phytocompounds and their application in the aquaculture industry.

Bioactivity	Plant Species	Phytocompound	Application	References
Antibacterial	*Chelidonium majus*	Chelerythrine chloride	Displayed strong toxicity against *Edwarsiella ictaluri* with a 24 h LC of 7.3 ± 0.8 mg/L and MIC of 2.1 ± 1.7 mg/L.	[125]
	*Eichhornia crassipes*	17-Pentatriacontene Octasiloxane Stigmasterol	Increased bacterial resistance in *Channa punctate* against *Vibrio harveyi* infection.	[126]
	*Gelsemium elegans*	Koumine Gelsemine	Significantly increased survival rates in *Megalobroma amblycephala* after the challenge with *Aeromonas hydrophila.*	[127]
	*Macleaya cordata*	Sanguinarine	Two concentrations (1 and 1.5 mg/kg of feed) improved the survival rate and resistance to *Vibrio parahaemolyticus* infection of *Litopenaeus vannamei.*	[128]
	*Platanus occidentalis*	Kaempferol 3-*O*-α-L-(2″, 3″-di-*Z*-*p*-coumaroyl)rhamnoside	High antibacterial efficacy against MRSA (IC_50_ 0.4 mg/L) and *Streptococcus iniae* LA94-426.	[129]
Antiparasitic	*Costus speciosus*	Gracillin Zingibernsis newsaponin	Can achieve 100% killing with in vitro treatments of gracillin and zingibernsis newsaponin. The EC50 values were 0.53 and 3.2 mg L^−1^, respectively.	[130]
Antiviral	*Galla chinensis*	Pentagalloylglucose	Elimination of all *Ichthyophthirius multifiliis* theronts at the concentrations of 2.5–20 mg/L, and complete interference of reproduction of tomonts at 40 mg/L. A 93.3% rate of survival was achieved in the Ich-infected catfish that were treated with pentagalloylglucose at 20 mg/L, whereas all infected fish were dead in the negative control group.	[131]
	*Macleaya microparpa*	Sanguinarine β-Allocryptopine 6-Methoxyl-dihydro-chelerythrine	Potent anthelmintic activity against *Dactylogyrus intermedius* in *Carassius auratus* with EC_50_ values of 0.37, 4.64 and 3.63 mg L^−1^, respectively.	[132]
	*Macleaya microparpa*	Dihydrosanguinarine Dihydrochelerythrin	The EC_50_ values of dihydrosanguinarine and dihydrochelerythrine against *I. multifiliis* were 5.18 and 9.43 mg/L, respectively.	[133]
	*Polygonum cuspidatum*	Emodin	At 96 min, in vitro treatment of emodin at 1 mg/L was able to kill all *I. multifiliis*. Recovery of the Ich-infected *Ctenopharyngodon idella* can be achieved by continuously adding emodin for 10 days.	[134]
	*Avicennia alba*	Friedlein Phytosterols 1-Triacontanol	Compounds friedlein, phytosterols and 1-triacontanol were determined to be potential drug candidates against WDSV using molecular docking simulation, with docking scores of −8.5 kcal/mol, −8.0 kcal/mol and −7.9 kcal/mol, respectively.	[135]
	*Gymnema sylvestre*	Gymnemagenol (3β, 16β, 28, 29-tetrahydroxyolean-12-ene)	At 20 µg/mL of gymnemagenol, it inhibited 50% of cell viability of grouper nervous necrosis virus (GNNV) that showed effectiveness in inhibiting the proliferation of GNNV in infected SIGE cells.	[136]
	*Polysiphonia morrowii*	3-Bromo-4,5-dihydroxybenzy methyl ether	3-Bromo-4, 5-dihydroxybenzy methyl ether exhibited significant antiviral activities showing selective index values (SI = CC_50_/EC_50_) of 20 to 40 against infectious hematopoietic necrosis virus (IHNV) and infectious pancreatic necrosis virus (IPNV).	[137]
	*Psidium guajava*	2,5-Bis(1,1-dimethylethyl) Diethyl phthalate Asarone Phthalic acid, butyl dodecyl ester Phytol	The F1-treated *Fennerropenaeus indicus* survived significantly against white shrimp syndrome virus (*p* < 0.05) at 80%, and while survival rates of 20, 30, 40, 35, 35 and 25 were found in the F2 to F7 fraction-treated groups, respectively. Meanwhile, the control group faced a 100% mortality rate.	[138]
	*Rhus verniciflua*	Fustin Fisetin Sulfurutin	Fisetin showed the highest significant anti-infectious activities against hemorrhagic necrosis virus and antiviral hemorrhagic septicemia virus, showing EC_50_ values of 27.1 and 33.3 µM. Fustin and sulfuretin displayed significant antiviral activities, showing EC_50_ values of 91.2–197.3 μM against infectious hemorrhagic necrosis virus and viral hemorrhagic septicemia virus.	[139]
Antifungal	*Eucalyptus camaldulensis*	Anethole 1,8-Cineole	*E. camaldolensis* at a concentration of 25 ppm and *G. herbarium* at a concentration of 100 ppm for 60 min daily with three repetitions were the best treatments in *Onchorynchus mykiss*, represented by the prevention of fungal attack, and the increase in the hatching rate, the eyed egg rate and the final larvae rate.	[140]
	*Geranium herbarium*	Geraniol Dihydrogeraniol		
	*Thymus linearis*	Hexadec-2-en-1-ol Carvacrol	More specificity of hexadec-2-en-1-ol towards the V-type ATPase site, and of carvacrol towards TKL protein kinase, of *Saprolegnia parasitica* responsible for the virulence of pathogens.	[141]
	*Ulva lactuta*	Palmitic acid Squalene	Displayed a significant zone of inhibition of antifungal activity against *Aspergillus niger* (36 mm).	[142]
	*Zataria multiflora*	p-Cymene Carvacrol	A concentration of 25 ppm for 60 min daily with three repetitions was the best treatment in *Onchorynchus mykiss*, represented by the prevention of fungal attack, and the increase in the hatching rate, the eyed egg rate and the final larvae rate.	[140]

## Data Availability

Not applicable.

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
