# Peer review of "Phytocompounds as an Alternative Antimicrobial Approach in Aquaculture"

_antibiotics, 2022, doi:10.3390/antibiotics11040469_

Round 1

Reviewer 1 Report

Although the intention of the review is good, it lacks some transcendental aspects to improve the relevance of the manuscript, in the same way, there is not a good relationship between the title and objectives of the work, with what is developed.

For example, and one of the most important aspects, the title indicates that phytocompounds will be discussed, however, few compounds are detailed in the manuscript, and in some cases they even mention activities of extracts (not compounds). In this sense, the manuscript should include a good review of different types of chemical compounds extracted from plant sources and that have antimicrobial activity.

On the other hand, the initial section of "Phytocompounds and their plant origins" only describes the chemical characteristics of the groups of compounds, but none in particular is described, much less, with anmicrobial activities. It seems to me that the description of the chemical characteristics of the compounds is unnecessary, or should be completed with examples of antimicrobial compounds and the plant of origin.

In conclusion, there is no review of chemical compounds with antimicrobial activity that have been obtained from plants.

Another aspect of caution is that in the development of the review it is not mentioned with sufficient emphasis what is the relationship of these antimicrobial alternatives with aquaculture, the benefits already demonstrated or works in which this use has already been experimentally analyzed.

Author Response

Reviewer comments

Reply

Reviewer #1

• The manuscript should include a good review of different types of chemical compounds extracted from plant sources and that have antimicrobial activity.

The amendment has been made to improve the title and the objectives of the work.

• The initial section of "Phytocompounds and their plant origins" only describes the chemical characteristics of the groups of compounds, but none in particular is described, much less, with anmicrobial activities.

The section “Phytocompounds and their plant origins” has been added with more discussions. The amendment has been made.
(Line 109-192)(Table 1-5).

• No review of chemical compounds with antimicrobial activity that have been obtained from plants.

The amendment has been made. (Line 316-478).

Reviewer 2 Report

The manuscript from Naqiuddin et al. discuss the perspective of using the plant-derived compounds as alternative for antimicrobial drugs in aquaculture. Overall, the manuscript is well presented and organized. For this reviewer, the authors could try to make the size of paragraphs more uniform. It could be accepted for publication after major revisions.

The authors should include the objective of the manuscript in the end of introduction section. They also could add a short description of the topics they discussed.

In the second section of the revision paper (Phytocompounds and their plant origins), the authors introduce the classes of phytocompounds. The section is well written and has the didactic explanation needed for a review manuscript. However, I missed a clear association of these section with aquaculture. It was too general in my opinion. I put some observation in the attached pdf file regarding the tables.  The authors could at least indicate in the respective tables the antimicrobial activity of each compound.

In the section ‘Extraction Methods of Phytochemical From Plants’ the authors could include a table showing the main advantages and disadvantages of each type of method.

On the other hand, the authors did not present any table for the section of ‘Phytocompound as alternatives of antimicrobial approach in aquaculture’ which is the main section of the manuscript. Please add a table summarizing this section.

Other suggestions are found in the attached pdf file.

Author Response

Reviewer comments

Reply

Reviewer #2

• The authors could try to make the size of paragraphs more uniform.

The amendment has been made.

• Line 99; The authors should include the objective of the manuscript in the end of introduction section.

The amendment has been made. The objective of the manuscript has been included (Line 102-106).

• Line 115; The text you put should be used as legend.

The amendment has been made (Line 128).

• Figure 1; Please move the figure

The amendment has been made.

• Table 1-5; The authors could at least indicate in the respective tables the antimicrobial activity of each compound.

The amendment has been made (Table 1-5).

• In the section ‘Extraction Methods of Phytochemical From Plants’ the authors could include a table showing the main advantages and disadvantages of each type of method.

The amendment showing the advantages and disadvantages has been made (Table 7) (Line 220-315).

• Line 283; Please include a table summarizing the compounds with antimicrobial activity.

The amendment has been made (Table 8) (Line 323).

Reviewer 3 Report

Abstract is poorly written and need to written again in good manner.

Introduction section should be rearranged again to be understandable. The good beginning should be related with the fish production and the aquaculture production, then the use of antibiotics and its hazards, then the important of natural products.

Why the author just mentioned Magnolia officinalis and Euphorbia fischeriana, despite there are thousands of medicinal plants.

 The aim of the current review not cleared and need to presented in adequate and scientific way.

 Each compound should be presented its beneficial effects in separated figure.

The presented information is so spherical. Each subtitle contain compound should discussed their effective role in depth.

 The author needs to explain why they addressed the Conventional Extraction Method (differences and its importance).

Table 7 should be presented the heat temperature and time and clarified is it direct or indirect heat.

Antibacterial, antifungal and antiviral activity mechanisms should be illustrated in separated figures.

Author Response

Reviewer comments

Reply

• Line 15-27; Abstract is poorly written and need to written again in good manner.

The amendment has been made (Line 17-29).

• Line 30-98; Introduction section should be rearranged again to be understandable. The good beginning should be related with the fish production and the aquaculture production, then the use of antibiotics and its hazards, then the important of natural products.

The amendment has been made. The introduction has been rearranged and the fish production and the aquaculture production was added (Line 32-101).

• Line 92-98; Why the author just mentioned Magnolia officinalis and Euphorbia fischeriana, despite there are thousands of medicinal plants.

The amendment has been made.

• Table 1-5; Each compound should be presented its beneficial effects in separated figure.

The amendment has been made (Table 1-5).

 • Table 1-5; Each subtitle contain compound should discussed their effective role in depth.

The amendment has been made (Table 1-5).

• The author needs to explain why they addressed the Conventional Extraction Method (differences and its importance)

The amendment has been made and added (Subsection 3.2) (Line 297-312) and Table 7 (Line 314).

• Table 7 should be presented the heat temperature and time and clarified is it direct or indirect heat.

The amendment has been made regarding the heat temperature and duration (Table 7) (Line 314).

• Antibacterial, antifungal and antiviral activity mechanisms should be illustrated in separately.

The amendment has been made. (Line 317-478).

Round 2

Reviewer 1 Report

The manuscript already has more robust information in accordance with the objectives of the document itself. The changes made are relevant and in accordance with previous suggestions.

Author Response

Dear Reviewer 1

Thank you for all the constructive feedbacks.

Reviewer 3 Report

The abstract section needs to devided into short sentences. Also, the main propose of the current review missed the objective that the author study benefits not hazards. Besides, the author s forget to addressed that they studying the extraction method. Overall, the abstract section need Great efforts to be suitable for publication.

The conclusion section need to addressed the main topics that had been covered by review.

Author Response

Reviewer comments

Rebuttal

Reviewer #2

The abstract section needs to devided into short sentences. Also, the main propose of the current review missed the objective that the author study benefits not hazards. Besides, the author s forget to addressed that they studying the extraction method. Overall, the abstract section need Great efforts to be suitable for publication.

The amendment has been made (Line 17-40).

The conclusion section need to addressed the main topics that had been covered by review.

The amendment has been made.  (Line 510-530).